# Imitation Beyond Expectation Using Pluralistic Stochastic Dominance

**Ali Farajzadeh, Danyal Saeed, Syed M. Abbas, Rushit Shah, Aadirupa Saha, Brian D. Ziebart**
Department of Computer Science
University of Illinois Chicago
Chicago, IL 60607
{afaraj5,dsaeed3,sabbas33,rshah231,aadirupa,bziebart}@uic.edu

## Abstract

Imitation learning seeks to estimate policies reflecting the values of demonstrated behaviors. Prevalent approaches learn to match or exceed the demonstrator's performance *in expectation* without knowing the demonstrator's reward function. Unfortunately, this does not induce pluralistic imitators that learn to support distinct demonstrations. We reformulate imitation learning using *stochastic dominance* over the demonstrations' reward distribution across a range of reward functions as our foundational aim. Our approach matches imitator policy samples (or support) with demonstrations using optimal transport theory to define an imitation learning objective over trajectory pairs. We demonstrate the benefits of pluralistic stochastic dominance (PSD) for imitation in both theory and practice.

## 1 Introduction

When learning from demonstrations, behaviors reflecting individual preferences and capabilities are often demonstrated. Existing imitation learning methods struggle to preserve these distinct behaviors while trying to improve beyond them. For example, inverse reinforcement learning (Abbeel & Ng, 2004; Ziebart, 2010) and discriminative imitation (Ratliff et al., 2006; Ho & Ermon, 2016) methods seek to match or outperform (Syed & Schapire, 2007) the demonstrations under a range of reward functions *in expectation*. As shown in Figure 1, this can be achieved by an imitator that never produces behavior that a demonstrator prefers over any of his or her more preferable demonstrations.

We seek a stronger *distributional* guarantee of **pluralistic stochastic dominance** (PSD)[1], which ensures the imitator a higher probability of achieving any level of reward than the demonstration distribution (i.e., stochastic dominance) for all reward functions (i.e., pluralism) within some defined set. This requires the imitator to *match or improve upon* the distinct properties of exceptional demonstrations rather than focusing on the average of demonstrations—often by randomizing between different modes (Figure 1). These guarantees support more complex applications of imitator policies (e.g., sampling many candidate trajectories and selecting the best) beyond the assumption that a single imitator trajectory is sampled and executed.

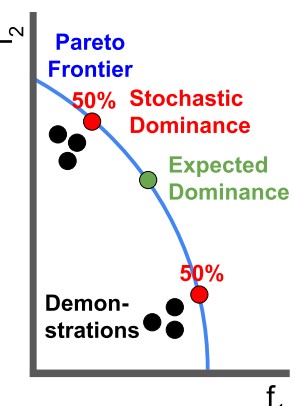

Figure 1: Dominance in expectation (green) guarantees better performance than the demonstration *average* for all conical sum reward functions. Pluralistic stochastic dominance (red) makes the *distribution of rewards* preferable by guaranteeing a higher probability of achieving any reward.

---

[1]Code available at https://github.com/Ali199776/PSD.

39th Conference on Neural Information Processing Systems (NeurIPS 2025).

Our approach is based on the observation that if all demonstrations can be matched to supported imitator behaviors that are better than the paired demonstration for all reward functions (Figure 1), then PSD is achieved (Armbruster & Luedtke, 2015). We employ optimal transport theory (Rioux et al., 2024) to perform this matching and a margin-based upper bound (Ziebart et al., 2022) to define a loss over the demonstration-imitation behavior pair. We then optimize the imitator policy using the matched pairs in the fully realizable and model-based imitator policy settings. We provide stochastic dominance generalization guarantees to establish the theoretical benefits of pursuing pluralistic stochastic dominance, and experimentally demonstrate better support for distinct behavior that improves beyond the paired demonstrations.

The main contributions of this paper are three-fold: First, PSD improves upon imitation learning methods designed for specific risk-sensitivities (Majumdar et al., 2017; Singh et al., 2018; Santara et al., 2018; Lacotte et al., 2019) to provide guarantees simultaneously across all reasonable risk-sensitive performance measures (e.g., value-at-risk, conditional value-at-risk, and range value-at-risk, detailed in Appendix A.1). These performance guarantees provide broader support for more complex use cases of imitation learning, such as selecting the best trajectory from a set of samples. Second, PSD-based imitation provides an alternative justification to maximum entropy inverse reinforcement learning for stochastic imitation policies. Finally, we introduce novel evaluation metrics for imitation learning based on stochastic and Pareto dominance that can be applied in settings with known reward bases, but not singular motivating reward function.

## 2 Background and related work

### 2.1 Inverse reinforcement learning

In imitation learning settings, reward functions defining desirable behavior are unknown. Instead, foundational formulations assume that reward features are available that define a linear (Ng & Russell, 2000; Abbeel & Ng, 2004) or conical sum (Syed & Schapire, 2007) reward function.

**Definition 2.1.** Given **reward features** $\mathbf{f} : \Xi \to \mathbb{R}^K$, for behavior trajectories $\xi \in \Xi$, the family of feature-based **reward functions** is defined as $r_\theta(\xi) = \theta \cdot \mathbf{f}(\xi)$ with parameters $\theta \in \mathbb{R}^K$ (**linear**) or $\theta \in \mathbb{R}_{\geq 0}^K$ (**conical sum**).

An imitation policy $\pi$ that matches the feature moments of the demonstrator guarantees equal rewards in expectation—including the demonstrator's unknown (linear) reward function (Abbeel & Ng, 2004):

$$\forall \theta \in \mathbb{R}^K, \mathbb{E}_{\xi \sim \mathbb{P}_\pi}\left[\mathbf{f}(\xi)\right] = \mathbb{E}_{\tilde{\xi} \sim \mathbb{P}_{\tilde{\pi}}}\left[\mathbf{f}(\tilde{\xi})\right] \implies \mathbb{E}_{\xi \sim \mathbb{P}_\pi}\left[r_\theta(\xi)\right] = \mathbb{E}_{\tilde{\xi} \sim \mathbb{P}_{\tilde{\pi}}}\left[r_\theta(\tilde{\xi})\right] \tag{1}$$

where $\mathbb{P}_\pi$ denotes the distribution over trajectories $\xi$ based on the interaction between the policy $\pi$ and the dynamics of the decision process, which we assume are deterministic.

Many imitation learning approaches can be viewed as matching various moments (rewards, on-policy/off-policy state-action value functions) of the demonstrator (Swamy et al., 2021), including: behavior cloning (Pomerleau, 1988); maximum margin planning (Ratliff et al., 2006); maximum entropy inverse reinforcement learning (Ziebart, 2010); DAGGER (Ross et al., 2011) generative adversarial imitation learning (Ho & Ermon, 2016); and Value Dice (Kostrikov et al., 2019).

Entropy regularization methods for reinforcement learning (Neu et al., 2017)—also known as softmax decision policies (Sutton & Barto, 2018)—increase the diversity of the imitator's trajectories within these moment-matching techniques. However, these provide robust *predictive guarantees* for imitation learning (Ziebart, 2010) rather than *performance guarantees* for diverse demonstrators. Extensions of these methods attempt to model variations in preferences or quality of demonstrations with latent variables. These are then used to condition policy models (e.g., as mixture models) or focus imitation on more desirable demonstrations (Brown et al., 2020b; Chen et al., 2021; Wu et al., 2019; Zhang et al., 2021). We aim to avoid the computational challenges (e.g., difficult nonconvex optimizations) and/or strong assumptions underlying these approaches.

### 2.2 Outperformance and subdominance minimization

Our approach is closer in motivation to methods designed to outperform demonstrators. Early methods focus on policies that outperform in terms of expected rewards (Definition 2.2).

**Definition 2.2.** Policy $\pi_1$ has **expected dominance** over $\pi_2$ if the expected trajectory reward under $\pi_1$ is at least as much as the expected trajectory reward under $\pi_2$: $\mathbb{E}_{\xi_1 \sim \pi_1} [\mathrm{r}_\theta(\xi_1)] \geq \mathbb{E}_{\xi_2 \sim \pi_2} [\mathrm{r}_\theta(\xi_2)]$ for fixed $\theta$.

MWAL (Syed & Schapire, 2007) and LPAL (Syed et al., 2008) guarantee outperforming the demonstrator in expectation under the assumption that the signs of the reward function weights are known (i.e., conical sum reward functions of Definition 2.1). In this setting, better expected reward features guarantee better expected rewards:

$$\forall \theta \in \mathbb{R}_{\geq 0}^K, \mathbb{E}_{\xi \sim \mathbb{P}_\pi} \left[ \mathbf{f}(\xi) \right] \succeq \mathbb{E}_{\tilde{\xi} \sim \mathbb{P}_{\tilde{\pi}}} \left[ \mathbf{f}(\tilde{\xi}) \right] \implies \mathbb{E}_{\xi \sim \mathbb{P}_\pi} \left[ \mathrm{r}_\theta(\xi) \right] \geq \mathbb{E}_{\tilde{\xi} \sim \mathbb{P}_{\tilde{\pi}}} \left[ \mathrm{r}_\theta(\tilde{\xi}) \right]. \tag{2}$$

Subdominance minimization (Ziebart et al., 2022) extends this idea of outperformance by seeking uniform dominance (Definition 2.3) across conical sum reward functions by minimizing a convex bound over the probability of violating uniform dominance.

**Definition 2.3.** Policy $\pi_1$ has **uniform dominance** over $\pi_2$ if all trajectory samples from $\pi_1$ have at least as much reward as all samples from $\pi_2$: $\mathbb{P}_{\xi_1 \sim \pi_1; \xi_2 \sim \pi_2}(\mathrm{r}_\theta(\xi_1) \geq \mathrm{r}_\theta(\xi_2)) = 1$ for fixed $\theta$.

Unfortunately, uniform dominance encourages deterministic policies that are similar to the expected dominance policy in settings like Figure 1.

We pursue a less strict notion of dominance in this paper: stochastic dominance (Definition 2.4). It is based on having a better distribution of rewards.

**Definition 2.4.** Policy $\pi_1$ has **stochastic dominance** over $\pi_2$ if $\pi_1$ has at least as much probability of exceeding any reward threshold: $\forall c \in \mathbb{R}, \mathbb{P}_{\xi_1 \sim \pi_1}(r_\theta(\xi_1) \geq c) \geq \mathbb{P}_{\xi_2 \sim \pi_2}(r_\theta(\xi_2) \geq c)$ for fixed $\theta$.

In terms of strictness, uniform dominance implies stochastic dominance, which implies expected dominance. However, uniform dominance is often infeasible (e.g., Figure 2a), while stochastic dominance is always feasible (e.g., $\pi = \tilde{\pi}$). In addition to expected reward bounds (2), stochastic dominance guarantees broad risk measure improvements (Ogryczak & Ruszczyński, 1999). We summarize some of these in Theorem 2.5.

**Theorem 2.5.** *Stochastic dominance of $r_\theta(\pi) \succeq r_\theta(\tilde{\pi})$ for some fixed $\theta$ guarantees improved expected and risk-sensitive rewards for the imitator $\xi \sim P_\pi$ with respect to the demonstrator $\tilde{\xi} \sim P_{\tilde{\pi}}$: $\forall (c \in [0,1], d \in (c,1]), \mathbb{E}_{\xi \sim \pi}[r_\theta(\xi)] \geq \mathbb{E}_{\tilde{\xi} \sim \tilde{\pi}}[r_\theta(\tilde{\xi})], VaR_c(r_\theta(\xi)) \geq VaR_c(r_\theta(\tilde{\xi})), CVaR_c(r_\theta(\xi)) \geq CVaR_c(r_\theta(\tilde{\xi})), and RVaR_{c,d}(r_\theta(\xi)) \geq RVaR_{c,d}(r_\theta(\tilde{\xi})).$*

Prior imitation learning research employs risk sensitivity narrowly to address safety concerns by targeting specific tail risks (and specific quantile levels). Extensions of generative-adversarial imitation learning (GAIL) (Ho & Ermon, 2016) match specific demonstrator risk-sensitivities (Majumdar et al., 2017; Santara et al., 2018; Lacotte et al., 2019). Bayesian estimation methods (Brown et al., 2020a; Javed et al., 2021) incorporate risk sensitivity to more robustly address uncertainty during reward function estimation, In contrast, we consider risk-sensitivity exhaustively—across a family of reward functions and over all sensitivity thresholds, as guaranteed by stochastic dominance (Theorem 2.5)—to incentivize high-quality coverage of diverse demonstrations.

## 2.3 Optimal transport

Optimal transport theory considers the minimum cost of transforming from one distribution, $\mathbb{P}_X$, to another $\mathbb{P}_Y$ under cost function $c : \mathcal{X} \times \mathcal{Y} \to \mathbb{R}_{\geq 0}$. The Kantorovich (1942) formulation defines this transformation using a joint probability measure $\gamma \in \Delta_{\mathcal{X} \times \mathcal{Y}}$ with $\gamma(x, y)$ representing the amount of probability mapped from $x$ to $y$ and marginals that match the source and target distributions. For discrete distributions (with $\mathbb{P}_X$ and $\mathbb{P}_Y$ supporting $m$ and $n$ values, respectively), this can be expressed as a linear program:

$$\mathrm{OT}_c(\mathbb{P}_X, \mathbb{P}_Y) = \min_{\gamma \geq \mathbf{0}} \sum_{i,j} \gamma_{i,j} c(x_i, y_j) \text{ s.t. } \forall j, \sum_i \gamma_{i,j} = \mathbb{P}_Y(y_j), \forall i, \sum_j \gamma_{i,j} = \mathbb{P}_X(x_i). \tag{3}$$

The optimization is solved exactly by the Hungarian algorithm in $\mathcal{O}(|\mathcal{X}|^3)$ time or with $\epsilon$ error tolerance using specialized algorithms (Dvurechensky et al., 2018) in $\tilde{\mathcal{O}}(\max(|\mathcal{X}|, |\mathcal{Y}|)^2/\epsilon^2)$ time.

The optimal transport objective (commonly referred to as the Wasserstein distance for metric costs) has been popularized as an alternative to the Jensen-Shannon divergence in generative-adversarial

learning (Arjovsky et al., 2017). Previous investigations for imitation learning tasks (Xiao et al., 2019; Dadashi et al., 2021). include cross-domain imitation transfer (Nguyen et al., 2021; Fickinger et al., 2022), combining trajectory matching with behavioral cloning (Haldar et al., 2023), and matching reward-less trajectories with expert trajectories (Luo et al., 2023). Each of these are distinct from our approach and motivation.

We build upon a key relationship between optimal transport and stochastic dominance in this work:

*Remark* 2.6. $\mathbb{P}_Y \succeq \mathbb{P}_X$ (Definition 2.4) if and only if there is a mapping from $\mathbb{P}_X$ to $\mathbb{P}_Y$ that is non-decreasing in value: $\mathrm{OT}_{\max(x-y,\,0)}(\mathbb{P}_X, \mathbb{P}_Y) = 0$, where $c(x, y)$ is positive only when $x > y$.

## 3   Approach

### 3.1   Pluralistic stochastic dominance

We consider imitation learning with multiple demonstrators. Each has their own reward function, $r_\theta$, presumed to be from the family of conical sum cost functions (Definition 2.1). Feature moment methods (1, 2) can guarantee that each demonstrator is at least indifferent between the trajectory distributions of the demonstrators and the imitator *in expectation* (Swamy et al., 2021). However, this does not guarantee any chance of producing highly desirable behavior for any demonstrators. As a consequence, if preferences are based on higher quantiles of reward distributions rather than expectations, the demonstration distribution can be highly preferable. We introduce pluralistic stochastic dominance (PSD) to ensure that the imitator policy is no less preferable to the demonstration distribution for all conical sum reward functions and all reward quantiles (Definition 3.1).

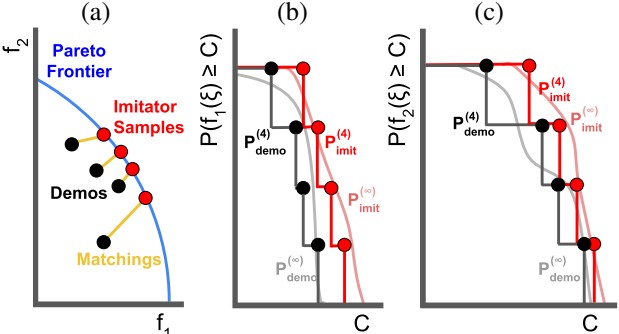

Figure 2: (a) Features ($f_1$, $f_2$) of four demonstrations (black points) matched (yellow lines) with four imitator samples (red points); the cumulative features for $f_1$ (b) and $f_2$ (c) for the sample distribution ($P^{(4)}$) and the full distribution ($P^{(\infty)}$). We seek to optimize the imitator policy using available demonstrator/imitator samples to achieve a full distribution for the imitator that stochastically dominates the demonstrator. This can be verified for each $\theta$ by sorting the samples, e.g., for $\theta = \begin{bmatrix} 1 \\ 0 \end{bmatrix}$ (b) and $\theta = \begin{bmatrix} 0 \\ 1 \end{bmatrix}$ (c). However, this is intractable for continuous sets of $\theta$; our upper bound instead employs a single matching (a).

**Definition 3.1.** Distribution $\mathbb{P}_\pi$ provides (first-order) **pluralistic stochastic dominance** over $\mathbb{P}_{\tilde{\pi}}$ (for all conical sum reward functions and for margin function $\beta_\theta$), which we denote $\mathbb{P}_\pi \succeq_{\mathrm{PSD}} \mathbb{P}_{\tilde{\pi}}$, iff:

$$\forall(\theta \in \mathbb{R}^K_{\geq 0}, c \in \mathbb{R}), \mathbb{P}_{\xi \sim \pi}(r_\theta(\xi) \geq c) \geq \mathbb{P}_{\tilde{\xi} \sim \tilde{\pi}}(r_\theta(\tilde{\xi}) \geq c) + \beta_\theta. \tag{4}$$

Though exact replication ($\pi = \tilde{\pi}$) trivially achieves this, imitation learning often aims to be performant on withheld demonstrations, so exact replication of finite training demonstrations is not sufficient. Instead, imitator policies $\pi$ that achieve better reward distributions (and better generalization) are desired. Figure 2(a) provides an example of strict stochastic dominance.

We extend stochastic dominance to the pluralistic setting by taking the maximum of optimal transport problems (Remark 2.6) for each conical sum reward function (Definition 3.2).

**Definition 3.2.** The **pluralistic stochastic subdominance** between imitator and demonstrator trajectory distributions (with margin function $\beta_\theta$) is given by the worst-case reward function:

$$\max_{\theta \in [0,1]^K} \overbrace{\left( \min_{\gamma \succeq \mathbf{0}} \sum_{i,j} \gamma_{i,j} \left[ r_\theta(\tilde{\xi}_j) - r_\theta(\xi_i) + \beta_\theta \right]_+ \text{ s.t. } \sum_j \gamma_{i,j} = \mathbb{P}_\pi(\xi_i) \; \forall i, \sum_i \gamma_{i,j} = \mathbb{P}_{\tilde{\pi}}(\tilde{\xi}_j) \; \forall j \right)}^{\mathrm{OT}_{[\Delta r_\theta + \beta_\theta]_+}(\mathbb{P}_\pi, \mathbb{P}_{\tilde{\pi}})}. \tag{5}$$

Minimizing this entire set of optimal transport problems to zero guarantees PSD (Theorem 3.3). The proofs of this theorem and others are provided in Appendix B.

**Theorem 3.3.** *Zero maximum optimal transport in Def. 3.2 (with $\beta_\theta = 0$) and pluralistic stochastic dominance are equivalent:* $\max_{\theta \in [0,1]^K} \mathrm{OT}_{[\Delta r_\theta]_+}(\mathbb{P}_\pi, \mathbb{P}_{\tilde{\pi}}) = 0 \iff \mathbb{P}_\pi \succeq_{\mathrm{PSD}} \mathbb{P}_{\tilde{\pi}}$.

As a result, PSD extends the risk-sensitive properties of stochastic dominance (Theorem 2.5) to the entire set of conical sum reward functions.

**Corollary 3.4.** *PSD guarantees that $\pi$ exhibits better risk-sensitive performance than $\tilde{\pi}$ under $r_\theta$ for the set of measures in Theorem 2.5 and all conical sum reward functions, i.e., $\theta \geq \mathbf{0}$.*

While the inner minimization (over $\gamma$) in (5), $\mathrm{OT}_{[\Delta r_\theta]_+}(\mathbb{P}_\pi, \mathbb{P}_{\tilde{\pi}})$, is a standard optimal transport linear program, the outer maximization (over $\theta$) is of a convex function (of $\theta$). This family of convex maximization programs is known to be NP-hard (Raghavachari, 1969), suggesting computational challenges for our specific instances, unfortunately (Dentcheva & Wolfhagen, 2015).

## 3.2 Matched subdominance minimization

Given the apparent computational challenges of exactly verifying pluralistic stochastic dominance (not only for Figure 2(b,c), but all $\theta \geq \mathbf{0}$), we instead derive a computationally efficient upper bound. We approach this by "pushing" the maximization of $\theta$ deeper into the original PSD expression:

$$\max_{\theta \in [0,1]^K} \min_{\substack{\gamma \succeq \mathbf{0} \text{ s.t.} \\ \sum_j \gamma_{i,j} = \mathbb{P}_\pi(\xi_i) \; \forall i \\ \sum_i \gamma_{i,j} = \mathbb{P}_{\tilde{\pi}}(\tilde{\xi}_j) \; \forall j}} \sum_{i,j} \gamma_{i,j} \underbrace{\left[ \mathrm{r}_\theta(\tilde{\xi}_j) - \mathrm{r}_\theta(\xi_i) + \theta \cdot \beta \right]_+}_{\mathrm{subdom}_{\mathbf{1},\beta}(\xi_i, \tilde{\xi}_j)}, \tag{6}$$

This replaces a set of optimal transport problems for each $\theta$ with one single optimal transport problem (Figure 2a) and makes numerous independent $\theta$ maximization problems that are easy to solve. Specifically, the resulting inner maximization of $\theta$ is equivalent to a particular instance of the **subdominance** (Ziebart et al., 2022) introduced for imitation learning via uniform dominance:

$$\mathrm{subdom}_{\boldsymbol{\alpha},\boldsymbol{\beta}}(\xi, \tilde{\xi}) = \sum_k \left[ \alpha_k \left( f_k(\tilde{\xi}) - f_k(\xi) \right) + \boldsymbol{\beta} \right]_+, \tag{7}$$

which measures how far trajectory $\xi$ is from Pareto-dominating $\tilde{\xi}$ (by a margin $\beta$ with features weighted by $\alpha$). We define our relaxed optimization problem as a linear program using the more general form of subdominance (Def. 3.5).

**Definition 3.5. Matched Subdominance Minimization** given $\boldsymbol{\alpha}$ and $\boldsymbol{\beta}$ is obtained from:

$$\mathrm{OT}_{\mathrm{subdom}_{\alpha,\beta}}(\mathbb{P}_\pi, \mathbb{P}_{\tilde{\pi}}) = \min_{\substack{\gamma \succeq \mathbf{0} \text{ s.t.} \\ \sum_j \gamma_{i,j} = \mathbb{P}_\pi(\xi_i) \; \forall i \\ \sum_i \gamma_{i,j} = \mathbb{P}_{\tilde{\pi}}(\tilde{\xi}_j) \; \forall j}} \sum_{i,j} \gamma_{i,j} \mathrm{subdom}_{\alpha,\beta}(\xi_i, \tilde{\xi}_j). \tag{8}$$

Margins $\beta > 0$ require strict improvement and avoid the trivial $\pi = \tilde{\pi}$ solution. The sets of trajectories should be sufficiently large to cover the distinct demonstrated behaviors. Given tens or hundreds of trajectories in each set, the optimal transport problem is not a critical computational bottleneck in practice.

As a result of being an upper bound, stochastic dominance can be guaranteed (Theorem 3.6).

**Theorem 3.6.** *For any $\alpha > 0$ and $\beta \geq 0$,*

$$\mathrm{OT}_{\mathrm{subdom}_{\alpha,\beta}}(\mathbb{P}_\pi, \mathbb{P}_{\tilde{\pi}}) = 0 \implies \mathbb{P}_\pi \succeq_{\mathrm{PSD}} \mathbb{P}_{\tilde{\pi}}.$$

We note that this is a special case of a recently established family of losses for which an optimal transport distance of zero implies multivariate stochastic dominance (Rioux et al., 2024).

## 3.3 Policy learning algorithms

We consider two imitation learning settings: **fully realizable** with any distribution over trajectories possible to learn; and **policy model** with a parametric policy model, $\pi_\theta$, optimized.

In the fully realizable setting, we consider a set of candidate trajectories, $\xi_i$, (ideally from the Pareto frontier) and learn the imitator's distribution over those trajectories (Def. 3.7).

**Definition 3.7.** For a given candidate set of trajectories, fixed $\alpha$ weights, and a distribution of demonstrations, the **matched minimal subdominance imitator policy** is obtained from:

$$\min_{\gamma \succeq 0} \sum_{i,j} \gamma_{i,j} \text{subdom}_{\alpha,\mathbf{1}}(\xi_i, \tilde{\xi}_j) + \lambda \text{Reg}(\gamma) \text{ such that: } \sum_i \gamma_{i,j} = \mathbb{P}_{\tilde{\pi}}(\tilde{\xi}_j) \ \forall j, \tag{9}$$

where regularizer $\text{Reg}(\gamma) = ||\gamma||$ or $\sum_i ||\gamma_{i,*}||$ encourages more uniform assignments and imitation trajectory distributions, respectively. The imitator trajectory distribution is then obtained by marginalizing: $\mathbb{P}_\pi(\xi_i) = \sum_j \gamma_{i,j}$.

Learned policy models enable generalization to different tasks within the same environment or to other environments. We leverage (deep) reinforcement learning methods (e.g., policy gradient optimization) using the subdominance-based optimal transport solution to determine a training signal for a policy model. This allows stochastically dominant policy optimization without first identifying a set of candidate trajectories (Def. 3.7). The model update procedure is described in Algorithm 1.

---

**Algorithm 1** Policy model update

**Input:** $M$ imitator samples $\{\xi_i\}$, $N$ demonstrations $\{\tilde{\xi}_j\}$, policy/parameters $\pi_\phi$, and learning rate $\eta$
**Output:** Updated policy/parameters $\pi_\phi$
1: Set $\mathbb{P}_\pi(\xi_i) = \frac{1}{M}$
2: Solve $\text{OT}_{\text{subdom}}$ given $\mathbb{P}_\pi(\xi_i)$ and $\mathbb{P}_{\tilde{\pi}}(\tilde{\xi}_j)$ (Def. 3.2)
3: Construct training signals $\{a_i\}$ from OT solution
4: Update model parameters $\phi$ using variables $\mathbf{a}$ from (10) or (11): $\phi \leftarrow \phi + \eta \sum_{i=1}^M a_i \nabla_\phi \log \mathbb{P}_\pi(\xi_i)$

---

Step 4 of the Algorithm parallels policy gradient methods (Williams, 1992) with $\{a_i\}$ replacing other improvement signals. These are obtained from the OT matching, $\gamma$, using **demonstration normalization** (10) or **weighted best match** (11), which emphasizes the best trajectories more:

$$a_i = \sum_j \gamma_{i,j} \left( \text{subdom}(\xi_i, \tilde{\xi}_j) - \sum_{i'} \gamma_{i',j} \, \text{subdom}(\xi_{i'}, \tilde{\xi}_j) \right), \tag{10}$$

$$a_i = \sum_j \left( \gamma_{i,j} \, \text{subdom}(\xi_i, \tilde{\xi}_j) - \mathbb{I}\left[ i = \text{argmin}_{i'} \, \text{subdom}(\xi_{i'}, \tilde{\xi}_j) \right] \sum_{i'} \gamma_{i',j} \, \text{subdom}(\xi_{i'}, \tilde{\xi}_j) \right). \tag{11}$$

Additionally, the $\alpha$ values of the subdominances for optimal transport (8) can either remain fixed, as implied by Algorithm 1, or be simultaneously updated using stochastic gradient optimization.

### 3.4 Generalization analysis

We characterize distributional stochastic dominance guarantees for the population of demonstrations based on a finite, IID training sample using the Dvoretzky et al. (1956) inequality.

**Theorem 3.8.** *Given the cumulative mass function (CMF) in the $K$-dimensional reward feature space from shifting the empirical demonstration CMF (with $N$ IID sampled trajectories):* $F_{\tilde{\pi}}^{N+}(\mathbf{f}) = \left[ F_{\tilde{\pi}}^N(\mathbf{f}) - \epsilon \right]_+ + \epsilon \mathbb{I}[\mathbf{f} = \infty]$, *and its corresponding probability mass function,* $\mathbb{P}_{\tilde{\pi}}^{N+}(\mathbf{f})$, *then:*

$$\text{OT}_{\text{subdom}_{\mathbf{1},0}}(\mathbb{P}_\pi, \mathbb{P}_{\tilde{\pi}}^{N+}) = 0 \implies \mathbb{P}\left( \forall \mathbf{f} \in \mathbb{R}^K \ F_\pi^N(\mathbf{f}) \le F_{\tilde{\pi}}^N(\mathbf{f}) \right) \ge 1 - NKe^{-2N\epsilon^2}.$$

This requires the convex hull of the Pareto frontier to be supported by a small number of points that each have at least $\epsilon$ imitator probability; and Pareto dominate at least $\epsilon$ of $\mathbb{P}_{\tilde{\pi}}^{N+}$. Cases in which $\mathbb{P}_{\tilde{\pi}}^{N+}(\mathbf{f})$ assigns probability to "unrealizable" features $\mathbf{f}$ are also addressed in Appendix B.

## 4 Experiments

### 4.1 Baseline imitators and evaluation metrics

As baseline methods for comparison, we evaluate: Maximum Entropy Inverse Reinforcement Learning (MaxEnt IRL) (Ziebart, 2010); Linear Programming Apprenticeship Learning (LPAL) (Syed et al., 2008); Generative Adversarial Imitation Learning (GAIL) (Ho & Ermon, 2016); an oracle version of InfoGAIL (Li et al., 2017) trained on pre-determined mode clusters (InfoGAIL*); Risk-Averse Imitation Learning (RAIL) (Santara et al., 2018); and regret-based Bayesian Robust Optimization for Imitation Learning (BROIL) (Brown et al., 2020a; Javed et al., 2021). Additional experimental

details are described in Appendix C. We use withheld testing demonstrations to evaluate the diversity and quality of imitator policies in two ways:

**Stochastic Dominance** estimates whether the imitation reward distribution is better than the demonstration reward distribution. We randomly select a set of weight vectors to induce various reward functions. For each weight vector, we evaluate whether the imitation policy stochastically dominates the testing demonstrations (Def. 2.4) and report the rate of stochastic dominance for each approach.

**Pareto Dominance** estimates when imitation trajectories are unambiguously better than demonstrations. We use the exact imitator policy or randomly sample a set of rollouts. We measure $\mathbb{P}(\mathbf{f}(\xi_{\mathrm{imit}}) \succeq \mathbf{f}(\tilde{\xi}_{\mathrm{demo}}))$ for each demonstration and report the minimum, average, and maximum.

## 4.2 Illustrative grid world experiments

We first consider `Lava World`, a deterministic grid environment from the robust imitation literature (Brown et al., 2020a). Each trajectory starts from the same initial state and seeks to reach a fixed goal state in the bottom-right corner of the grid. At each time step, the agent can move in any of the four cardinal directions. Trajectories are characterized by two features: the number of white and red cells traversed (Figure 3). The cost of a trajectory is computed as a weighted sum of these features. A trajectory terminates either when the agent arrives at the goal state or when a fixed time horizon (e.g., 10) is reached.

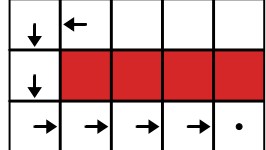

Figure 3: Sample demonstration in the `Lava World` grid environment with white and red (lava) grid cells.

While the cost feature weights for this environment are unknown, we are provided with a set of demonstrated trajectories. To train and evaluate our approach, we first divide a set of trajectories (with unique features) reaching the goal within 10 timesteps into imitator candidates (when on or near the Pareto frontier) and demonstrations (when less optimal). We then further divide the demonstrations into two random, equally-sized subsets for training and testing.

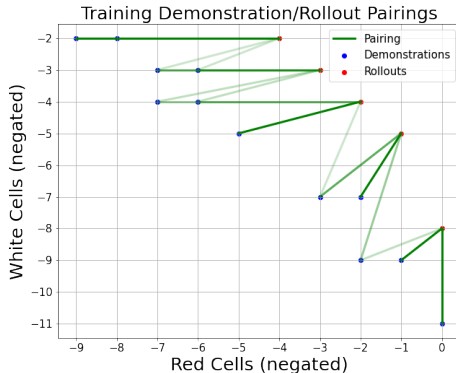
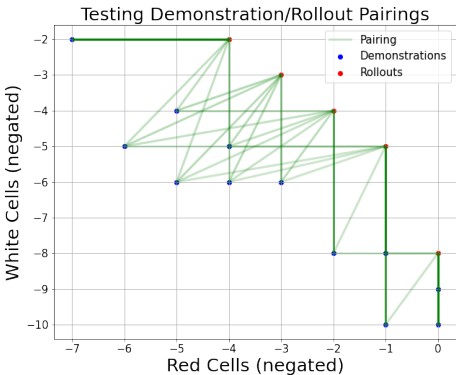

Figure 4: For training (left), demonstrations (blue) are paired with rollouts (red) via (9) to construct the imitator distribution. For testing (right), withheld demonstrators are paired with that imitator distribution via (8). Darker green pairings correspond to larger $\gamma$ values.

We employ our fully-realizable training approach (Definition 3.7) for PSD. We first prune the candidate trajectory set by removing trajectories that are Pareto-dominated by others in the set. This ensures that only trajectories with potentially optimal rewards remain to define the imitator's policy. Subsequently, we match the training set demonstrations with the pruned candidate trajectories by solving a quadratic program based on Eq. (9) with fixed subdominance variables $\alpha = 1$ and $\beta = 0.5$, and $L_2$ regularization of the imitator trajectory distribution $\mathbb{P}(\xi)$ to promote greater uniformity over the set of candidates, resulting in improved generalization to unseen demonstrations. Figure 4 (left) shows this matching for a particular training sample.

To verify generalized stochastic dominance after training, withheld demonstrations are matched to the imitator's trajectory distribution using Equation (8). If the objective of the matching problem

is zero (equivalently, the imitation trajectories all dominate their paired demonstration trajectories), then stochastic dominance is guaranteed, as Figure 4 (right) shows.

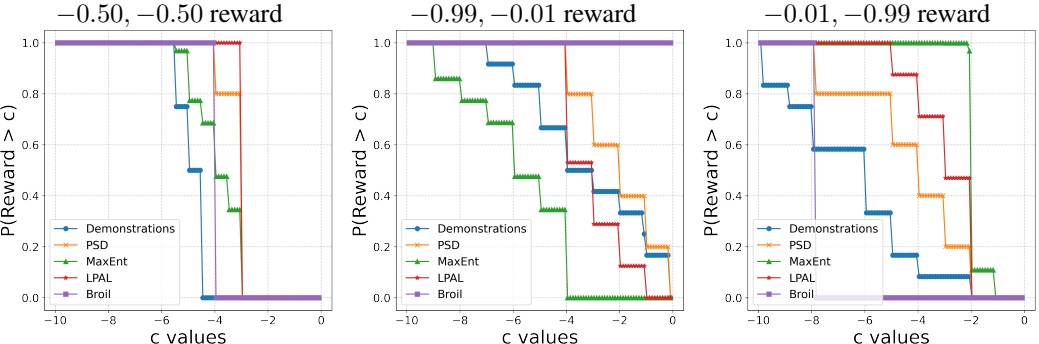

Figure 5: Cumulative rewards of the trajectory distributions of the demonstrations, PSD, MaxEnt IRL, and Regret methods for three reward functions.

Figure 5 shows the excess reward distributions for three reward functions. Other imitation methods produce curves that are worse for some portions of some reward functions, reflecting the trade-offs they make for better performance in other portions of these curves. In contrast, PSD produces excess reward curves that are strictly better than the demonstration curves for all three reward functions on the train-test split of Figure 4, indicating better mode coverage.

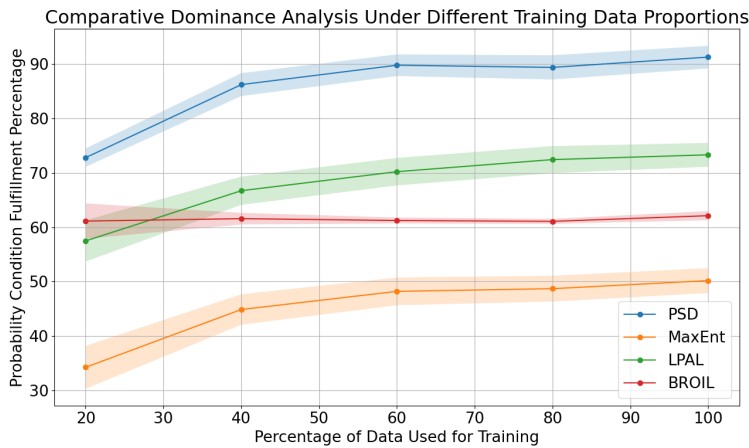

Figure 6: Average dominance (with standard error) for different training data amounts.

Figure 6 shows the percentage of randomly produced reward functions for which each imitation learning method stochastically dominates the withheld demonstration set, and how it changes with the amount of training demonstrations. PSD provides a higher rate of stochastic dominance across the entire range of training data sizes compared to other methods.

Table 1: Frequency of Pareto dominance over demonstrations in `Lava World`.

| Policy | min | avg | max |
|---|---|---|---|
| Demonstrations | **0.08** | 0.16 | 0.50 |
| MaxEnt IRL | 0.00 | 0.33 | **1.00** |
| LPAL | 0.00 | 0.46 | **1.00** |
| BROIL | 0.00 | 0.15 | **1.00** |
| PSD | **0.08** | **0.47** | 0.92 |

Table 1 provides summary statistics for how frequently demonstrations are Pareto dominated by the imitator's trajectory distribution (or withheld demonstrations as an additional baseline). PSD Pareto-dominates all demonstrations with at least $8\%$ probability. In contrast, all other imitators fail to have any probability of Pareto dominating at least one demonstration. Additionally, PSD provides the highest average Pareto dominance, with nearly half of the imitator trajectories dominating (i.e., being unambiguously better) than the demonstrations.

## 4.3 Policy model optimization

Our second set of experiments considers policy model optimization (Alg. 1) in the `Point Bot` (Javed et al., 2021) and `Reacher` (Todorov et al., 2012) environments. `Point Bot` is a continuous robotic task for navigating a point mass (subject to noisy, velocity-based air resistance) in a two-dimensional plane from a starting position to a pre-defined and stationary goal, ostensibly without passing through a gray region (obstacles). The robot moves by applying a force in a cardinal direction. `Reacher` is a robotic arm with two rigid links and two joints (Figure 7). The end of one link is fixed to the center of the environment. In our multi-modal variant, the goal is to move the robot's end effector to one of two targets (red or yellow) by applying appropriate sequences of torques.

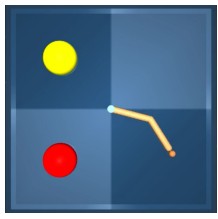

Figure 7: `Reacher` environment.

For the `Point Bot` environment, the imitators learn from the set of human-demonstrated trajectories shown in Figure 8 (left). Some demonstrations completely avoid the obstacles (gray areas), others partially avoid obstacles, while many appear entirely oblivious of obstacles. The trajectories are characterized by the number of timesteps in gray areas, the number of timesteps in white areas, and the sum of distances to the goal location over the trajectory. To facilitate multi-modal policy learning, we increase the number of layers of the policy model from two to four (each with 64 fully connected hidden nodes) compared to prior work (Javed et al., 2021). For PSD, we train this model using the *demonstration normalization* variant of Algorithm 1.

| (a) Demonstrations | (b) Behavior Cloning | (c) GAIL | (d) InfoGAIL* | (e) PSD |
|---|---|---|---|---|

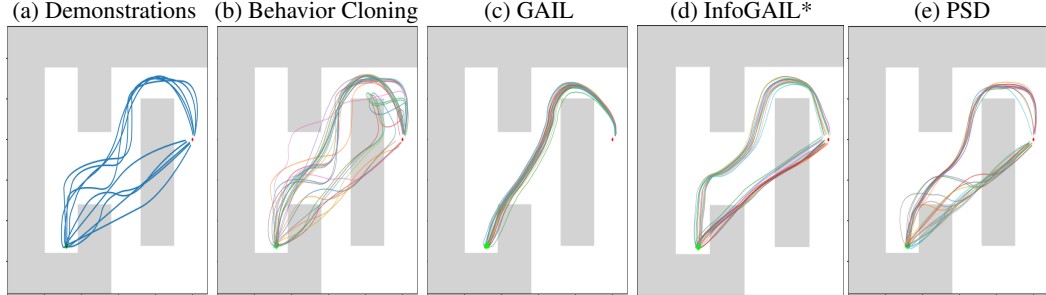

Figure 8: `Point Bot` training demonstrations starting from a lower left origin and moving to the goal in the upper right (a); and sample trajectories from policies learned using each method (b)-(e).

Behavior cloning produces trajectories that are similar to the demonstrations, but often of lower quality. Specifically, the sample trajectories in Figure 8b encounter more obstacles and sometimes fail to efficiently reach the goal. We initialize the other imitation learning methods from this behavior cloning policy as a starting point. GAIL (Figure 8c) suffers from mode collapse, ultimately producing trajectories that all encounter obstacles. InfoGAIL* (Figure 8d) is given a binary mode membership (obstacle-avoiding or obstacle-oblivious) of each trajectory and learns a separate model for each mode. By matching each demonstrated mode's means, the resulting trajectories tend to be suboptimal compared with the best demonstrations of the corresponding mode. PSD imitation (Figure 8e) produces trajectories that are of higher quality while still covering the spectrum of trade-offs between grays cells and white cells/distance of the demonstrations. Significant support is provided to these two modes: obstacle avoidance and obstacle obliviousness. However, some trajectories also cover trade-offs between the two modes (e.g., avoiding the first obstacle, but not the second). This illustrates the flexibility of PSD to operate in settings without clearly specified latent spaces of modes (e.g., binary-valued modes).

For the `Reacher` environment, demonstrations are synthetically produced using reinforcement learning—more specifically, the soft actor-critic (SAC) algorithm (Haarnoja et al., 2018)—to learn policies for two different targets, the yellow circle and the red circle. A SAC-learned policy (for the yellow or the red goal) and a uniformly random policy are sampled with complementary probabilities to produce the trajectories of the demonstration set with an equal number of red goal trajectories and yellow goal trajectories. The features we incorporate are the sum of distances from each of the targets over the entire trajectory. The policy model that the imitators train is a Gaussian multi-layer perceptron with four layers of 64 neurons. For PSD, we train this policy model using the *weighted best match* variant of Algorithm 1 and also report PSD with optimized $\alpha$ parameters (PSD-$\alpha^*$), as described in §3.3.

Table 2: Frequency of imitator Pareto and stochastic dominance of demonstrations.

| | Point Bot | | | | Reacher | | | |
| | Pareto | | | Stochastic | Pareto | | | Stochastic |
| Policy | min | avg | max | avg | min | avg | max | avg |
|---|---|---|---|---|---|---|---|---|
| Demonstrations | 0.000 | 0.222 | 0.444 | 0.000 | 0.000 | 0.144 | 0.333 | 0.000 |
| Behavior Cloning | 0.001 | 0.180 | 0.353 | 0.000 | 0.001 | 0.154 | 0.290 | 0.000 |
| GAIL | 0.000 | 0.015 | 0.031 | 0.002 | 0.000 | 0.005 | 0.023 | 0.425 |
| RAIL | 0.000 | 0.004 | 0.031 | 0.000 | 0.000 | 0.000 | 0.000 | 0.000 |
| InfoGAIL$^*$ | 0.000 | 0.227 | 0.496 | 0.000 | 0.409 | 0.474 | 0.500 | 0.071 |
| PSD | 0.070 | 0.326 | 0.493 | 0.420 | 0.452 | 0.498 | **0.547** | 0.561 |
| PSD-$\alpha^*$ | **0.080** | **0.387** | **0.642** | **0.657** | **0.466** | **0.500** | 0.534 | **0.662** |

Table 2 provides statistics for how well demonstrations are supported by the imitator policy (Pareto dominance) using 1000 policy rollouts, and whether the reward distributions of the imitator are strictly better than the demonstration reward distribution (stochastic dominance) averaged over 1000 random reward functions for both `Point Bot` and `Reacher`. Similarly to the fully realizable experiments, there are some demonstrations that are very difficult for the baseline methods to outperform (near zero minimum Pareto dominance values). The `Point Bot` demonstrations pose a significant challenge because some are distinct from the two main demonstrated modes, causing InfoGAIL* to also perform poorly in terms of minimum Pareto dominance. In contrast, the PSD policy produces trajectories that provide coverage of all demonstrations. More broadly, PSD excels across all metrics because it tends to produce trajectories that are: **in proportion** with the demonstrated behavior modes and often of **higher quality** than the demonstrations comprising that mode. PSD's high frequency of stochastic dominance illustrates the effectiveness of our policy gradient optimization in achieving the PSD objective. The other imitation methods are unfortunately unable to maintain the modes of the demonstrations and generally exhibit poor performance across all of these metrics, with a few exceptions, as a result.

## 5 Discussion and conclusions

This paper introduces stochastic dominance as an important property of distributional alignment (Sorensen et al., 2024) for imitation learning when demonstrations reflect the differing preferences of distinct demonstrators. Stochastic dominance provides stronger guarantees for demonstrators than expectation-matching imitation methods: reward distributions for each demonstrator that are at least as good as the demonstrated distribution—despite not knowing each demonstrator's exact reward function—for all common risk-sensitive measures. This avoids policies that are compromises between competing objectives by maintaining stochasticity. Though directly achieving stochastic dominance appears computationally difficult, we establish a relaxation using optimal transport theory that leads to exact algorithms in the fully-realizable setting and policy gradient algorithms when training a policy model. Through qualitative and quantitative analyses we show that our imitation learning approach provides support to all demonstrated behavior modes, while aiming to produce better quality behavior within those modes, leveraging concepts of both Pareto and stochastic dominance.

There are multiple important directions for future research. We have focused on deterministic dynamics in this paper. While our policy model optimizations naturally extend to stochastic environments, additional analyses and experimental validation remain as future work. Next, though hand-engineered reward features are reasonable for engineered systems (e.g., self-driving vehicles, robotics), many imitation learning methods learn reward functions without such features being available. Integrating reward feature learning in a manner that leverages potential multi-modality of demonstrations using our framework is an important future direction to avoid the limitation of known reward features. Finally, one key challenge is that policy optimization based on distributional criteria appears more challenging than maximizing expected rewards. Exploration of both on-policy and off-policy reinforcement learning in this context is likely needed for scaling to larger environments.

## Acknowledgments

This work was supported by the National Science Foundation under award #2312955.

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

# A   Additional Background Material

## A.1   Risk-sensitivity

Limiting the probability and magnitude of undesirable outcomes is crucial in many applications. Popular metrics for assessing these include *value at risk* (Def. A.1), *conditional value at risk* (Def. A.2), and *range value at risk* (Def. A.3).

**Definition A.1.** The **Value-at-Risk (VaR)** for a random reward variable $X$ and given quantile $c \in [0, 1]$, is the inverse of the cumulative density function $F(X)$ (if it exists) (12) or the generalized inverse (e.g., for discrete variables) (13):

$$\text{VaR}_c(X) = \nu_c(X) = \begin{cases} F_X^{-1}(c) & (12) \\ \inf_x \{x \mid \mathbb{P}(X \leq x) \geq c\}. & (13) \end{cases}$$

**Definition A.2.** The **Conditional Value-at-Risk (CVaR)**, measures the expected value within the tail of a reward distribution for a given quantile $\alpha$: $\text{CVaR}_c(X) = \mathbb{E}_X[X | X \leq \nu_c(X)]$.

**Definition A.3.** The **Range Value-at-Risk (RVaR)** discards both tails (defined by $c$ and $d$) of the reward distribution and computes the expectation for the range in between: $\text{RVaR}_{c,d} = \mathbb{E}_X[X \mid \nu_c(X) \leq X \leq \nu_d(X)]$.

## A.2   Stochastic Dominance

We review three multivariate extensions of stochastic dominance (Definition 2.4).

**Definition A.4** (Distribution-based stochastic dominance (Kopa & Petrová, 2018))**.** Random variable **X stochastically dominates (in distribution)** random variable **Y** *if and only if* $\forall \mathbf{c} \in \mathbb{R}^K \ P(\mathbf{X} \preceq \mathbf{c}) \leq P(\mathbf{Y} \preceq \mathbf{c})$.

**Definition A.5** (Utility-based stochastic dominance (Armbruster & Luedtke, 2015))**.** Random variable **X stochastically dominates** random variable **Y** *if and only if* $\mathbb{E}[u(\mathbf{X})] \geq \mathbb{E}[u(\mathbf{Y})]$ for any non-decreasing utility function $u : \mathbb{R} \to \mathbb{R}$.

**Definition A.6** (Linear stochastic dominance (Dentcheva & Wolfhagen, 2015))**.** Random variable **X stochastically dominates (in distribution)** random variable **Y** *if and only if* $\forall \theta \geq \mathbf{0}$, $\theta \cdot \mathbf{X}$ has univariate stochastic dominance over $\theta \cdot \mathbf{Y}$.

The linear stochastic dominance definition motivates our primal formulation using the traditional imitation learning assumption of linear reward functions Abbeel & Ng (2004). However, the utility-based definition is the strongest of these three (Armbruster & Luedtke, 2015; Kopa & Petrová, 2018). From an imitation learning perspective, it guarantees that a demonstrator or user with an unknown non-decreasing utility function will prefer the distribution of random rewards from variable $X$ over the distribution of random rewards from variable $Y$. Our dual relaxation aligns with this definition.

## A.3   Optimal Transport

We provide a proof sketch for Remark 2.6, which establishes a direct correspondence between stochastic dominance and optimal transport. For simplicity, we assume a one-to-one mapping between distributions, but the argument easily extends when the mapping is not one-to-one.

*Proof sketch for Remark 2.6.* The (generalized) inverse $\nu_c$ of Definition A.1, provides notation for defining a mapping between distributions for $X$ and $Y$, namely using the pairing $(\nu_c(X), \nu_c(Y)) \ \forall c \in [0, 1]$, which maps between values at the same quantile. If $Y$ stochastically dominates $X$, then $\nu_c(X) \leq \nu_c(Y) \ \forall c \in [0, 1]$. This corresponds to an optimal transport solution with value 0 for cost functions that are 0 when $x \leq y$.

In the other direction, if the optimal transport solution has a value of 0, then for some complete set of pairings $(c_1, c_2)$, $\nu_{c_1}(X) \leq \nu_{c_2}(Y)$. If $c_1 \neq c_2$ for some of these pairings, then given one pairing $(c_a, c_x)$ such that $c_a > c_x$, there must be another pairing $(c_b, c_y)$ such that $c_b < c_y$ and $c_b < c_a$ by the pigeonhole principle, resulting in $\nu_{c_b}(X) \leq \nu_{c_a}(X) \leq \nu_{c_y}(Y) \leq \nu_{c_x}(Y)$. Re-pairing these as $(c_a, c_y)$ and $(c_b, c_x)$ does not increase any cost as an optimal transport solution since $\nu_{c_a}(X) \leq \nu_{c_y}(Y)$ and $\nu_{c_b}(X) \leq \nu_{c_x}(Y)$, and brings the pairing closer to sorted matching. This re-pairing procedure can be continued until the resulting pairing is exactly $(\nu_c(X), \nu_c(Y)) \ \forall c \in [0, 1]$. This pairing is exactly the definition of stochastic dominance (Definition 2.4). □

# B  Proofs of Theorems and Lemmas

*Proof of Theorem 3.3.* We first prove that $\max_{\theta \in [0,1]^K} \mathrm{OT}_{[\Delta r_\theta]_+}(\mathbb{P}_\pi, \mathbb{P}_{\tilde{\pi}}) = 0 \implies P_\pi(r_\theta(\xi)) \succeq_{\mathrm{PSD}} P_{\tilde{\pi}}(r_\theta(\tilde{\xi}))$. First, consider the optimal assignment for a specific $\theta \in [0,1]^K$ in Definition 3.2: $\gamma^*(\theta)$. Since the corresponding objective value is assumed to be zero, this implies that:

$$\forall \gamma_{i,j}(\theta)^* > 0, r_\theta(\xi_i) \geq r_\theta(\tilde{\xi}_j) \tag{14}$$

$$\implies \gamma_{i,j}(\theta)^* \mathbb{I}\left[r_\theta(\tilde{\xi}_j) \geq C\right] = \gamma_{i,j}(\theta)^* \mathbb{I}\left[r_\theta(\tilde{\xi}_j) \geq C\right] \times \mathbb{I}\left[r_\theta(\xi_i) \geq C\right], \forall C \in \mathbb{R}, \tag{15}$$

where $\mathbb{I}[x]$ is an indicator function that evaluates to 1 if expression $x$ is true and 0 otherwise. We leverage this equality (15) in (b) below to prove the implication:

$$\forall (C, \theta) \in \mathbb{R} \times [0,1]^K, \quad \mathbb{P}_\pi(r_\theta(\xi) \geq C) \triangleq \sum_{i,j} \gamma_{i,j}^*(\theta) \mathbb{I}\left[r_\theta(\xi_i) \geq C\right] \tag{16}$$

$$\overset{(a)}{\geq} \sum_{i,j} \gamma_{i,j}^*(\theta) \mathbb{I}\left[r_\theta(\xi_i) \geq C\right] \times \mathbb{I}\left[r_\theta(\tilde{\xi}_j) \geq C\right]$$

$$\overset{(b)}{=} \sum_{i,j} \gamma_{i,j}^*(\theta) \mathbb{I}\left[r_\theta(\tilde{\xi}_j) \geq C\right]$$

$$\triangleq \mathbb{P}_{\tilde{\pi}}(r_\theta(\tilde{\xi}) \geq C),$$

where inequality (a) results from adding an additional condition. This inequality is then generalized to all positive values of $\theta$:

$$\forall (\alpha, \theta) \in \mathbb{R}_{\geq 0} \times [0,1]^K, \ \mathbb{P}_\pi(\alpha r_\theta(\xi) \geq \alpha C) \geq \mathbb{P}_{\tilde{\pi}}(\alpha r_\theta(\tilde{\xi}) \geq \alpha C)$$

$$\implies \forall \theta \in \mathbb{R}_{\geq 0}^K, \ \mathbb{P}_\pi(r_\theta(\xi) \geq C) \geq \mathbb{P}_{\tilde{\pi}}(r_\theta(\tilde{\xi}) \geq C).$$

In the other direction $(P_\pi(r_\theta(\xi)) \succeq_{\mathrm{PSD}} P_{\tilde{\pi}}(r_\theta(\tilde{\xi})) \implies \max_{\theta \in [0,1]^K} \mathrm{OT}_{[\Delta r_\theta]_+}(\mathbb{P}_\pi, \mathbb{P}_{\tilde{\pi}}) = 0)$, our proof is constructive. For any $\theta \in \mathbb{R}_{\geq 0}^K$, sort the trajectories $\xi_i$ and $\tilde{\xi}_j$ according to $r_\theta(\cdot)$. Then choose $\gamma_{i,j}^*(\theta)$ that matches $\xi_i$ and $\tilde{\xi}_j$ according to the sorted order with the weight based on the remaining unmatched probabilities of $\mathbb{P}_\pi(\xi_i)$ and $\mathbb{P}_{\tilde{\xi}}(\tilde{\xi}_j)$. Since $\mathbb{P}_\pi(r_\theta(\xi) \geq C) \geq \mathbb{P}_{\tilde{\pi}}(r_\theta(\tilde{\xi}) \geq C)$, then $\gamma_{i,j} > 0 \implies [r_\theta(\tilde{\xi}_j) - r_\theta(\xi_i)]_+ = 0$, so $\mathrm{OT}_{[\Delta r_\theta]_+}(\mathbb{P}_\pi, \mathbb{P}_{\tilde{\pi}}) = 0$. This holds for all $\theta \in [0,1]^K$. $\square$

*Proof of Theorem 2.5.* Leveraging the results of Theorem 3.3, we consider the $\gamma_{i,j}(\theta)$ for which $\sum_{i,j} \gamma_{i,j} \left[r_\theta(\tilde{\xi}_j) - r_\theta(\xi_i)\right]_+ = 0$. This implies that:

$$\forall (i,j), \gamma_{i,j}(\theta)\, r(\xi_{i)}) \geq \gamma_{i,j}(\theta)\, r_\theta(\tilde{\xi}_j) \tag{17}$$

$$\implies \sum_{i,j} \gamma_{i,j}(\theta)\, r(\xi_{i)}) \geq \sum_{i,j} \gamma_{i,j}(\theta)\, r_\theta(\tilde{\xi}_j) \tag{18}$$

$$\implies \mathbb{E}_{\xi \sim \mathbb{P}_\pi}\left[r(\xi)\right] \geq \mathbb{E}_{\tilde{\xi} \sim \mathbb{P}_\pi}\left[r_\theta(\tilde{\xi})\right]. \tag{19}$$

Further, using the definition of pluralistic stochastic dominance (Definition 3.1), we have: $\mathbb{P}_\pi \succeq_{\mathrm{PSD}} \mathbb{P}_{\tilde{\pi}} \iff \forall \theta \in \mathbb{R}_{\geq \mathbf{0}}^K, C \in \mathbb{R}$:

$$\mathbb{P}_\pi \succeq_{\mathrm{PSD}} \mathbb{P}_{\tilde{\pi}} \text{ iff: } \forall (\theta \in \mathbb{R}_{\geq \mathbf{0}}^K, C \in \mathbb{R}), \tag{20}$$

$$\mathbb{P}_\pi(r_\theta(\xi) \geq C) \geq \mathbb{P}_{\tilde{\pi}}(r_\theta(\tilde{\xi}) \geq C) \tag{21}$$

$$\mathbb{P}_\pi(r_\theta(\xi) \leq C) \leq \mathbb{P}_{\tilde{\pi}}(r_\theta(\tilde{\xi}) \leq C). \tag{22}$$

For convenience, we define these probabilities as $p$ and $q$:

$$p = \mathbb{P}_\pi(r_\theta(\xi) \leq C) \leq \mathbb{P}_{\tilde{\pi}}(r_\theta(\tilde{\xi}) \leq C) = q \tag{23}$$

$$\nu_p(r_\theta(\xi)) = C \qquad \text{and} \qquad \nu_q(r_\theta(\tilde{\xi})) = C \tag{24}$$

but $p \leq q$ and as VaR monotonically increases with increasing confidence level (if $\alpha' \geq \alpha$ then $\nu_{\alpha'}(X) \geq \nu_\alpha(X)$):

$$\nu_q(\mathbf{r}_\theta(\xi)) = C' \geq C \tag{25}$$

As pluralistic stochastic dominance holds for all $C$, VaR guarantee holds for all $\alpha$

$$\therefore, \quad \text{VaR}_\alpha(\mathbf{r}_\theta(\xi)) \geq \text{VaR}_\alpha(\mathbf{r}_\theta(\tilde{\xi})) \qquad \forall \alpha \in [0, 1]. \tag{26}$$

The proof for CVaR trivially follows from (26). If a function is always smaller than the other, its definite integral and average (with restricted domain) will also be smaller:

$$\rho_\alpha(\mathbf{r}_\theta(\xi)) = \frac{1}{\alpha} \int_0^\alpha \nu_\gamma(\mathbf{r}_\theta(\xi)) \, d\gamma \geq \frac{1}{\alpha} \int_0^\alpha \nu_\gamma(\mathbf{r}_\theta(\tilde{\xi})) \, d\gamma = \rho_\alpha(\mathbf{r}_\theta(\tilde{\xi})) \tag{27}$$

$$\text{CVaR}_\alpha(\mathbf{r}_\theta(\xi)) \geq \text{CVaR}_\alpha(\mathbf{r}_\theta(\tilde{\xi})) \qquad \forall \alpha \in [0, 1]. \tag{28}$$

And for the same reason, if we have different limits of the definite integral, the inequality still holds:

$$\eta_{\alpha,\beta}(\mathbf{r}_\theta(\xi)) = \frac{1}{\beta - \alpha} \int_\alpha^\beta \nu_\gamma(\mathbf{r}_\theta(\xi)) \, d\gamma \geq \frac{1}{\beta - \alpha} \int_\alpha^\beta \nu_\gamma(\mathbf{r}_\theta(\tilde{\xi})) \, d\gamma = \eta_{\alpha,\beta}(\mathbf{r}_\theta(\tilde{\xi})) \tag{29}$$

$$\text{RVaR}_{\alpha,\beta}(\mathbf{r}_\theta(\xi)) \geq \text{RVaR}_{\alpha,\beta}(\mathbf{r}_\theta(\tilde{\xi})) \qquad 0 \leq \alpha \leq \beta \leq 1. \tag{30}$$

$\square$

**Lemma B.1.** *The matched subdominance minimization value (Def. 3.5) upper bounds the worst-case reward difference optimal transport value:* $OT_{subdom_{\mathbf{1},\mathbf{0}}}(\mathbb{P}_\pi, \mathbb{P}_{\tilde{\pi}}) \geq \max_{\theta \in [0,1]} OT_{[\Delta r_\theta]_+}(\mathbb{P}_\pi, \mathbb{P}_{\tilde{\pi}}).$

*Proof.* Starting from Definition 3.2:

$$\max_{\theta \in [\mathbf{0},\mathbf{1}]} \min_{\gamma \succeq 0} \sum_{i,j} \gamma_{i,j} \left[ \theta \cdot \mathbf{f}(\tilde{\xi}_j) - \theta \cdot \mathbf{f}(\xi_i) \right]_+ \overset{(a)}{\leq} \min_{\gamma \succeq 0} \max_{\theta \in [\mathbf{0},\mathbf{1}]} \sum_{i,j} \gamma_{i,j} \left[ \theta \cdot \mathbf{f}(\tilde{\xi}_j) - \theta \cdot \mathbf{f}(\xi_i) \right]_+ \tag{31}$$

$$\overset{(b)}{\leq} \min_{\gamma \succeq 0} \sum_{i,j} \gamma_{i,j} \max_{\theta \in [\mathbf{0},\mathbf{1}]} \left[ \theta \cdot \mathbf{f}(\tilde{\xi}_j) - \theta \cdot \mathbf{f}(\xi_i) \right]_+ \tag{32}$$

$$\overset{(c)}{=} \min_{\gamma \succeq 0} \sum_{i,j} \gamma_{i,j} \text{subdom}_{\mathbf{1},\mathbf{0}}(\xi_i, \tilde{\xi}_j) \tag{33}$$

$$\overset{(d)}{\leq} \min_{\gamma \succeq 0} \sum_{i,j} \gamma_{i,j} \text{subdom}_{\mathbf{1},\boldsymbol{\beta}}(\xi_i, \tilde{\xi}_j),$$

where: $(a)$ follows from the maxmin-minmax inequality; (b) makes the maximizing choice of $\theta$ independently for each pair $(i, j)$ with $\gamma_{i,j} > 0$; (c) results from the subdominance being the worst-case difference in rewards for the imitator; and (d) is nondecreasing as the subdominance margin increases. $\square$

*Proof of Theorem 3.6.* Since the optimal matched subdominance (Definition 3.5) upper bounds the optimal pluralistic risk-sensitive matching (Definition 3.2) via Lemma B.1, an objective value of zero for the former implies an objective value of zero for the latter. Theorem 3.3 then implies pluralistic stochastic dominance. $\square$

Figure 9 shows when minimizing the matched subdominance is unnecessary for pluralistic stochastic dominance (i.e., $\mathbb{P}_\pi \succeq_{\text{PSD}} \mathbb{P}_{\tilde{\pi}} \not\Longrightarrow OT_{subdom_{\mathbf{1},\mathbf{0}}}(\mathbb{P}_\pi, \mathbb{P}_{\tilde{\pi}}) = 0$).

*Proof of Theorem 3.8.* The right-sided multivariate Dvoretzky-Kiefer-Wolfowitz inequality (Dvoretzky et al., 1956; Naaman, 2021) provides probabilistic bounds on the deviation between the empirical multivariate cumulative mass function (CMF) with $N$ samples ($F_{\tilde{\pi}}^N$) and the true population CMF ($F$) from the right as (See proof of Lemma 4.1 in (Naaman, 2021)):

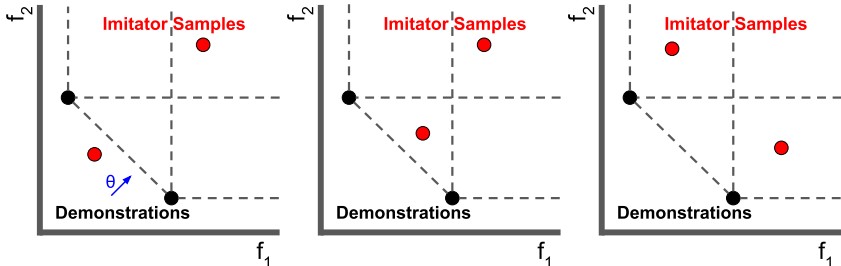

Figure 9: (left) Imitation not being a pluralistic stochastic dominator of demonstrations (and the $\theta$ maximizing the improvement violation); (center) imitation being a pluralistic stochastic dominator, but paired subdominance is nonzero; (right) imitation being a pluralistic stochastic dominator and paired subdominance is zero.

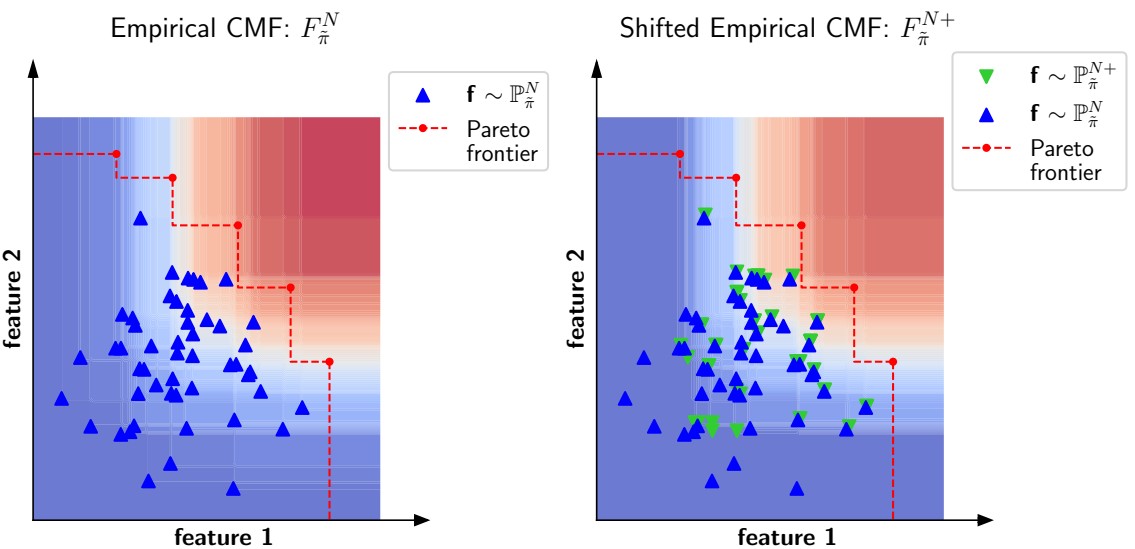

Figure 10: (Left) A sample population of demonstration feature vectors (in blue) in two dimensions overlaying the heatmap of its empirical CMF, and (Right) the heatmap of the empirical CMF shifted to the right by $\epsilon = 0.05$, along with feature vector samples from its corresponding PMF obtained via finite differences (in green).

$$\mathbb{P}\left(D_n^+ > \epsilon\right) = \mathbb{P}\left(\sup_{\mathbf{f}\in\mathbb{R}^K}\left(F_{\tilde{\pi}}^N(\mathbf{f}) - F(\mathbf{f})\right) > \epsilon\right) \le NKe^{-2N\epsilon^2}, \tag{34}$$

which is the same as saying

$$\mathbb{P}\left(\sup_{\mathbf{f}\in\mathbb{R}^K}\left(F_{\tilde{\pi}}^N(\mathbf{f}) - F(\mathbf{f})\right) \le \epsilon\right) \ge 1 - NKe^{-2N\epsilon^2} \tag{35}$$

since $\sup_{\mathbf{f}\in\mathbb{R}^K}\left(F_{\tilde{\pi}}^N(\mathbf{f}) - F(\mathbf{f})\right) \le \epsilon \iff \forall \mathbf{f} \in \mathbb{R}^K, \left(F_{\tilde{\pi}}^N(\mathbf{f}) - F(\mathbf{f})\right) \le \epsilon$, we have:

$$\mathbb{P}\left(\left(F_{\tilde{\pi}}^N(\mathbf{f}) - \epsilon\right) \le F(\mathbf{f})\right) \ge 1 - NKe^{-2N\epsilon^2} \ \forall \mathbf{f} \in \mathbb{R}^K \tag{36}$$

Our approach considers the worst-case distribution within these bounds and characterizes when stochastic dominance of the worst-case can be guaranteed. This perspective is inspired by entropic analysis under these bounds (Learned-Miller & DeStefano, 2008).

Consider the worst-case CMF based on this bound:

$$F_{\tilde{\pi}}^{N+}(\mathbf{f}) = \left[F_{\tilde{\pi}}^N(\mathbf{f}) - \epsilon\right]_+ + \epsilon\mathbb{I}[\mathbf{f} = \infty], \tag{37}$$

where $\epsilon$ is reduced from every $\mathbf{f} \in \mathbb{R}^K$ and added to the largest possible value of $\mathbf{f}$. The corresponding probability mass function, $\mathbb{P}_{\tilde{\pi}}^{N+}(\mathbf{f})$ may be obtained from the CMF via finite differences.

If $\mathbb{P}_{\tilde{\pi}}^{N+}(\mathbf{f}) > 0$, but if $\mathbf{f}$ is not Pareto-dominated by any realizable demonstrator trajectory, we call $\mathbf{f}$ *unrealizable*. Let $u \triangleq \mathbb{P}_{\tilde{\pi}}^{+}(\mathbf{f} \text{ unrealizable})$ denote the probability under $\mathbb{P}_{\tilde{\pi}}^{N+}$ of cost features that are not possible to Pareto-dominate (or equal) by any realizable demonstrator trajectory. The sum of these unrealizable probabilities is an integer multiple of $\epsilon$, to which we add one more $\epsilon$ (the one subtracted from the CMF at every input value). We assume that these unrealizable probabilities $u$ get assigned, or "matched", to the worst-case realizable demonstrator trajectory $\xi_{\text{wc}}$ during optimal transport. See Figure 10 for an example of the discussed quantities in two dimensions with $N = 50$ and $\epsilon = 0.05$ for the case when $u = 0$.

From (36), we see that the random variable with CMF $F_{\tilde{\pi}}^{N+}(\mathbf{f})$ stochastically dominates the random variable with CMF $F$ by a margin of $\epsilon$ with probability at least $(1 - NKe^{-2N\epsilon^2})$.

Now if $\max_{\xi_{\text{wc}}} \text{OT}(\mathbb{P}_\pi - u\delta_{\xi_{\text{wc}}}, \mathbb{P}_{\tilde{\pi}}^{N+} - u\delta_{\xi_{\text{wc}}}) = 0$ for the remaining distributions after making the worst-case assignment of unrealizable probability to $\xi_{\text{wc}}$, in other words, if our approach achieves pluralistic stochastic dominance over the trajectory distribution of the worst-case CMF, i.e., $\mathbb{P}_\pi \succeq_{\text{PSD}} \mathbb{P}_{\tilde{\pi}}^{N+}$ (by Theorem 3.6), then by (36) we have pluralistic stochastic dominance over the demonstrator *population* trajectory distribution $\mathbb{P}_{\tilde{\pi}}$ with probability at least $(1 - NKe^{-2N\epsilon^2})$, i.e., $\mathbb{P}(\mathbb{P}_\pi \succeq_{\text{PSD}} \mathbb{P}_{\tilde{\pi}}) \geq (1 - NKe^{-2N\epsilon^2})$. $\qquad\square$

## C   Additional Experimental Details

We provide additional experimental details in this section, including expanded interpretations of our evaluation measures (Section C.1), implementation details (Section C.2), and supplementary experimental evaluations (Section C.3).

### C.1   Evaluation Measure Interpretations

Since assessing stochastic dominance across the entire set of conical sum reward functions directly is computationally challenging (Section 3.1), we instead assess the dominance of the imitator over the demonstrator from two different perspectives.

Our **stochastic dominance** measure considers the frequency of stochastic dominance over randomly sampled cost functions rather than the guarantee for the entire set of reward functions. For each sample reward function, stochastic dominance guarantees:

- That for all non-decreasing utility functions applied to the sampled reward function, the imitator is preferable to the demonstrations (Def A.5); and

- For any reward threshold, the imitator has an equal or higher probability of exceeding that threshold compared to the demonstration distribution (Def 2.4).

Our **Pareto dominance** evaluation measure considers uniform dominance (Definition 2.3) between sampled imitator trajectories and the demonstration set of trajectories, indicating how frequently the imitator trajectory is preferred for all conical sum reward functions. Large values across the entire set of samples indicate good alignment with the demonstrations. Low values do not prevent stochastic dominance (as shown in Figure 9, center), but they tend to indicate some degree of misalignment. We report the minimum, average, and maximum of these samples to assess how well the imitator trajectory samples align with demonstrations.

### C.2   Implementation Details

Our implementation builds upon OpenAI Spinning Up[2], PG-BROIL (Javed et al., 2021)[3], and BROIL (Brown et al., 2020a)[4] repositories.

---

[2]https://github.com/openai/spinningup
[3]https://github.com/zaynahjaved/pg-broil
[4]https://github.com/dsbrown1331/broil

For Lava world experiments, for reporting the results comparing different approaches with different amounts of training data and frequency of imitator Pareto dominance, we have randomly split the whole set containing 24 demonstrations into two equal splits of training and testing 100 times, and the represented results are averaged. We have used the time horizon of 10 for trajectories. For solving the Quadratic Program (QP), we have used MOSEK optimizer, and have set the regularization parameter $\lambda$ to 0.001. For subdominance calculation during training, we have set the $\beta$ parameter to 0.5. The threshold for achieving the goal in `Point Bot` was originally in the default setting of 1, however, we have increased it to 10. The initial features for this environment are the number of gray and white cells, however, we added another feature that takes into account the distance from the target.

We use REINFORCE algorithm for policy optimization. For our policy network, we have used a Gaussian Multi-Layer Perceptron (MLP) with 4 hidden layers each having 64 neurons with the Tanh activation function. The network receives the agent's observations and produces a mean and standard deviation for each action dimension, and the agent takes actions by sampling from this Gaussian distribution. For optimization, we used the Adam optimizer with a learning rate of $3e - 5$, and for the subdominance calculation, we set the $\beta$ parameter to 0.001. We solve the QP using MOSEK optimizer. Training goes on for 2000 iterations, and the best model is saved according to the lowest QP objective value. We use 10 demonstrations for training, they come from two main modes, with each mode having 5 demonstrations. During each iteration, we rollout 30 trajectories. For PSD-$\alpha^*$, we have used Adam optimizer with a learning rate of $5e - 4$ for learning alpha values. Alpha values are initialized uniformly, sum to 1, and are always at least equal to 0.1. Training goes on for 4000 iterations and similarly the best model is saved.

The `reacher` environment we use has two targets with fixed positions, one labeled with a yellow circle and the other red. We use two sets of demonstrations, each containing 15 trajectories, for two different modes of behavior. The modes include moving the robot's end effector to the yellow or red circle. For Reacher, we have used the same policy optimization algorithm we used for `Point bot`, REINFORCE. The policy network architecture and QP optimizer are also the same. We have used Adam optimizer with a learning rate of $3e$-3, a $\beta$ parameter value of $0.001$, and with the MOSEK regularization parameter $\lambda$ set to 1000. During each iteration, we rollout 60 trajectories. Training goes on for 400 iterations and the best model is saved based on the lowest QP objective value. For PSD-$\alpha^*$, we have used Adam optimizer with a learning rate of $1e - 2$ for learning alpha values. Similar to `Point Bot`, alpha values are also initialized uniformly, sum to 1, and are at least 0.1. Training goes on for 1000 iterations and similarly the best model is saved.

For `Point Bot`, baseline experiments (GAIL, RAIL, InfoGAIL, BC) were run on several different personal computers and the slowest one took less than 5 hours to converge (e.g. on a laptop with 2.6GHz 10-core CPU, 32GB RAM). For `Reacher`, GAIL seemed to converge much earlier than `Pointbot`, taking close to an hour to converge (showing minimal improvements with longer training). RAIL was trained for 2-3 hours on `Reacher` but failed to display any improvement in our metrics. Experiments for PSD were run on an in-house server with GPU acceleration (equipped with two Nvidia GTX 1080 Ti GPUs), taking close to 1 hour and 1.5 hours each for convergence with `Pointbot` and `Reacher`, respectively.

## C.3 Supplemental evaluations

We provide additional experimental results in this section.

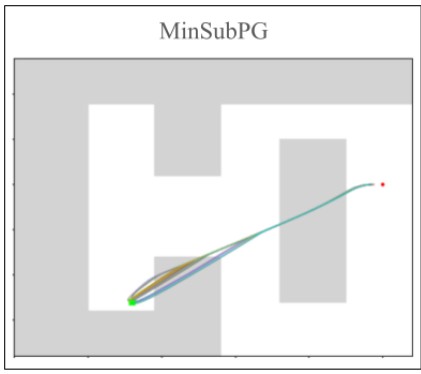

Figure 11: Trajectory samples from a policy learned using subdominance minimization (Ziebart et al., 2022).

Figure 11 shows the resulting trajectory samples from a policy learned using uniform subdominance minimization (Ziebart et al., 2022) in the `Point Bot` experiment of Section 4.3. Like GAIL (Ho & Ermon, 2016) (Figure 8(c)), it converges to a single mode of behavior. In this case, the mode is a "compromise" between the values reflected by the demonstrations. Unfortunately, since other modes are ignored, the resulting imitation policy does not provide good performance guarantees relative to the entire demonstration distribution.

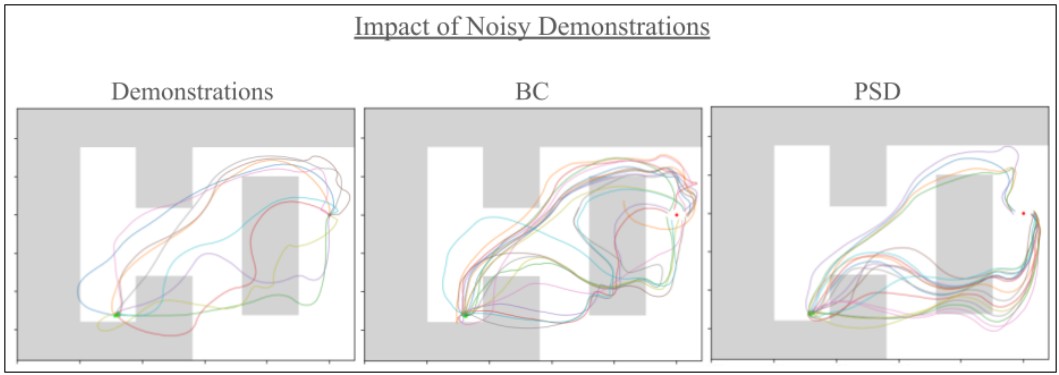

Figure 12: Noisy demonstration trajectories (left), trajectory samples from a behavioral-cloned policy (center) and trajectory samples from policy learned using PSD (right).

We next investigate imitation learning using a "noisier" set of demonstrations (Figure 12, left). Learning from these noisier demonstrations exacerbates the suboptimalities of behavior cloning (Figure 12, center), with far greater amounts of trajectory in the gray portions of the environment for many of the trajectories compared to the demonstrations (Figure 12, left) and to behavior cloning from less noisy demonstrations (Figure 8b). The PSD policy (Figure 12, right) produces better trade-offs of distance and obstacle with its trajectories, though with noticeable suboptimality compared to the PSD policy learned from less noisy data (Figure 8e). This suggests that incorporating a larger margin may be needed when learning from noisier demonstrations.

