# OpenReview forum: "Imitation Beyond Expectation Using Pluralistic Stochastic Dominance"
_NeurIPS.cc/2025/Conference — NeurIPS 2025 spotlight_

### Official Review · Reviewer_X6Xw · 2025-06-29

**Clarity:** 4
**Significance:** 3
**Originality:** 4
**Rating:** 5
**Confidence:** 3

**Summary:**

This paper identifies and addresses a key deficiency in imitation learning (IL): the tendency for algorithms to learn policies that match the *average* performance of demonstrators, potentially leading to behaviors that are a compromise between different expert strategies and are not truly preferred by any single demonstrator. The authors propose a more robust and intuitive objective: **Pluralistic Stochastic Dominance (PSD)**. The goal is to learn an imitator policy whose distribution of rewards is provably better than the demonstrators' distribution, not just in expectation, but across all reward quantiles and for an entire class of plausible reward functions (conical sum rewards).

To make this concept practical, the work forges a strong connection between PSD and optimal transport (OT). It shows that achieving PSD is equivalent to minimizing a worst-case OT cost. Since solving this directly is intractable, the authors derive a computationally efficient, convex upper bound they term "Matched Subdominance Minimization," which can be solved as a linear program. The paper presents algorithms for both a tabular "fully realizable" setting and a deep reinforcement learning setting where the objective serves as a loss for policy gradient updates. The experimental results on several environments convincingly show that the PSD approach successfully preserves distinct behavioral modes from the demonstration set while improving upon them, outperforming a suite of strong IL baselines on novel and well-suited evaluation metrics.

**Questions:**

1. A point of clarification regarding the implementation details for the fully-realizable setting. Line 268 mentions that the training demonstrations are matched with candidate trajectories by "solving a quadratic program based on Eq. (10)". However, there is no Equation (10) in the paper. Is this referring to the optimization problem in Definition 3.7 (line 203), perhaps with the L2 regularization on the transport plan `γ` making it a quadratic program?

2. Second, concerning the policy model optimization (Algorithm 1), the OT problem is solved in the inner loop to generate training signals. Could the authors comment on the practical computational overhead of this step, especially as the number of demonstration $N$ and imitator $M$ samples grows?

**Ethical Concerns:**

["NO or VERY MINOR ethics concerns only"]

**Final Justification:**

This paper introduces a novel and insightful method for imitation learning that goes beyond fitting the average of demonstrations. The derivation and analysis are rigorous, and properly supported by carefully designed experiments. This work receives positive feedback from all reviewers.

Thus I keep my rating at 5 - Accept

**Limitations:**

The choice of experimental environments is logical and serves to clearly illustrate the core properties of the PSD framework. However, to fully establish the scalability and robustness of the policy optimization algorithm, it would be compelling to see results on more standard, high-dimensional continuous control benchmarks. For instance, tasks from the MuJoCo suite (e.g., Humanoid) would challenge the policy optimization with more complex dynamics and action spaces, providing stronger evidence of its general applicability in deep RL settings.

Furthermore, the true power of pluralistic imitation would be most apparent on large-scale, heterogeneous datasets that are becoming standard in offline RL and IL research. While the paper's experiments use synthetically generated demonstrations with clear modes, real-world datasets are often more "messy." It would be highly impactful to evaluate PSD on such data. For instance, the **D4RL benchmark** contains mixed-quality data from diverse sources (e.g., scripted policies, human players, and partially trained agents), which is a perfect use-case for a method designed to improve upon, but not average over, demonstrations. More recent efforts like **MimicGen** or the **IKEA ASM** datasets provide demonstrations from multiple humans with inherent variability in strategy and skill. Testing PSD in these settings would provide powerful evidence of its ability to handle the complexity of real-world demonstration data, moving beyond the clean modes in the current experiments and truly validating its "pluralistic" promise.

Fu, J., Kumar, A., Nachum, O., Tucker, G., & Levine, S. (2020). D4rl: Datasets for deep data-driven reinforcement learning. arXiv preprint arXiv:2004.07219.

Mandlekar, A., et al. (2023). Mimicgen: A data generation system for scalable robot learning using human demonstrations. arXiv preprint arXiv:2310.17596.

Ben-Shabat, Y., Yu, X., Saleh, F., Campbell, D., Rodriguez-Opazo, C., Li, H., & Gould, S. (2021). The ikea asm dataset: Understanding people assembling furniture through actions, objects and pose. In Proceedings of the IEEE/CVF Winter Conference on Applications of Computer Vision (pp. 847-859).

**Quality:**

3

**Strengths And Weaknesses:**

#### Strength

This shift from matching moments to dominating distributions is a significant conceptual contribution to the field.

The theoretical development is rigorous and elegant. The authors do an excellent job of connecting the high-level goal of PSD to a practical algorithmic solution through the lens of optimal transport. The relaxation from the intractable worst-case OT problem to the tractable "Matched Subdominance Minimization" is clever and provides the key that makes this framework usable. The presentation is exceptionally clear, with Figures 1 and 2 being particularly effective at providing intuition for the core concepts. The experimental validation is also a high point; the authors not only compare against relevant and strong baselines but also propose new evaluation metrics (stochastic and Pareto dominance rates) that directly measure what the paper claims to achieve, which is a hallmark of thoughtful research.

#### Weaknesses

Despite these significant strengths, the work has a few limitations that warrant discussion. The most critical is the framework's reliance on pre-defined, hand-engineered reward features. This assumption, while common in the classic IRL literature, is a major departure from many modern IL approaches (like GAIL) that can learn directly from high-dimensional states (e.g., pixels). This limits the immediate applicability of the method to domains where such features are not readily available. The authors rightly acknowledge this as an area for future work. Additionally, the methodology relies on solving an optimal transport problem in the learning loop, which may present scalability challenges for very large numbers of demonstration and imitator trajectories compared to methods that rely on simpler adversarial training.

---

> ### Author Rebuttal · Authors · 2025-07-30
>
> Thank you for your careful reading of our paper and insightful questions.
>
> Weakness #1–Available cost features: We agree that requiring cost features limits the applicability of our approach compared to methods like GAIL or diffusion-based techniques. However, we also note that in many “real” applications of imitation learning for engineered systems (e.g., autonomous vehicles, robotics), cost features are available and widely employed.
>
>
> Question #1–Missing quadratic equation: Thank you for pointing out this error in the writing. You are indeed correct that this is meant to refer to the equation in Def. 3.7, and L2 regularization yields a QP.
>
> Question #2–Optimal transport complexity: For our experiments (n=10, m=30), we use an off-the-shelf QP solver (MOSEK), which has negligible run time (<20ms) compared to the m=30 rollouts. In theory, for uniform distributions, the Hungarian algorithm is O(m^3) time for the exact solution, but we find MOSEK to scale at O(mn) in practice from m*n of 10^3 to 10^6. If applied at larger scales, specialized algorithms with ε error rate guarantees require Õ(mn/ε^2) time complexity (Dvurechensky et al. 2018). However, we would favor taking randomized subsets of the demonstrations (for very large n) as training batches to avoid this potential computational bottleneck, assuming the true set of distinct modes is only moderately large. Importantly, apart from the O(mnk) complexity of computing the pairwise costs, none of these complexities depends on the cost feature set size (k).
>
> We appreciate the pointers to environments/datasets that are well-aligned with the motivations of PSD and plan to investigate these in future work. We will aim to alleviate concerns about OT being a bottleneck based on your concerns.

---

### Official Review · Reviewer_EQ92 · 2025-07-02

**Clarity:** 3
**Significance:** 3
**Originality:** 3
**Rating:** 4
**Confidence:** 2

**Summary:**

The paper proposes an approach to imitation learning by introducing Pluralistic Stochastic Dominance (PSD) as an alternative to traditional expectation-based or risk-sensitive objectives. Rather than matching average performance, PSD ensures that the learned policy provides a better distribution over reward outcomes than the demonstrator for all conical sum reward functions. This is achieved via an optimal transport-based formulation that matches demonstration and imitation trajectories while enforcing a form of reward dominance.
The paper provides a theoretical formulation of PSD and its connection to stochastic dominance, a practical relaxation of the intractable optimization using matched subdominance minimization. The paper also presents learning algorithms for both fully realizable and policy-model-based settings and a suite of empirical evaluations (grid world, Point Bot, Reacher) showing that PSD-based methods outperform or match baseline approaches in both Pareto and stochastic dominance metrics.

**Questions:**

1. How does the approach handle or adapt to situations where no explicit reward features are available, as in high-dimensional visual domains? Would it be feasible to integrate learned representations into the PSD framework?

2. While the outer maximization over reward weights is convex, it is noted to be NP-hard. Can the authors provide more insights into how this scales in practice for high-dimensional feature spaces?

3. The method relies on a subdominance margin (e.g., β = 0.5 in experiments). How sensitive are the results to this choice, and what guidance can be offered for tuning this parameter?

4. InfoGAIL* is included as a strong baseline. Could the authors provide further discussion on whether PSD could be augmented with latent variable models for additional gains, or does PSD render such modeling unnecessary?

5. Since policy optimization relies on subdominance-derived signals rather than reward gradients, how stable is learning over time? Any failure cases or instability patterns observed?

Evaluation Criteria Shift:
A rebuttal addressing questions 1, 2, and 3, especially regarding how PSD could generalize beyond known reward features and scale computationally, would significantly increase confidence in broader applicability and lead to a higher score.

**Ethical Concerns:**

["NO or VERY MINOR ethics concerns only"]

**Final Justification:**

I believe my questions have been addressed, I confirm my score, and I have completed the mandatory acknowledgement.

**Limitations:**

The paper explicitly discusses core limitations:
- it assumes access to known reward features (limiting generalization).
- current methods are restricted to deterministic dynamics; extending to stochastic settings is left for future work.
- the computational hardness of exact PSD verification is acknowledged and addressed via approximations.
Further discussion of sample efficiency, especially under high-dimensional state/action spaces, and the feasibility of deployment in complex real-world environments would improve the section.

**Paper Formatting Concerns:**

No concerns.

**Quality:**

3

**Strengths And Weaknesses:**

Quality: Strong theoretical grounding: PSD is rigorously defined and connected to prior work in optimal transport and stochastic dominance. Derivation of tractable approximations via convex optimization is well-justified and novel. Experiments are well-controlled and convincingly demonstrate the benefits of the approach across multiple settings and metrics.
While the theory is sound, the computational burden of the proposed methods is only partially addressed, particularly the outer maximization over reward functions. The policy optimization algorithm relies on handcrafted reward features, which limits applicability in more general, high-dimensional settings.

Clarity: The paper is clearly written and well-organized, with useful diagrams (e.g., Figures 1, 2, 4, and 8) illustrating core concepts. The step-by-step exposition of theoretical constructs, optimization strategy, and experimental setup is effective.
Some key terms (e.g., “supporting behaviors”, “matched subdominance”) are introduced with heavy notation before intuitive descriptions are provided. The transition from full PSD definitions to the practical relaxation could be clarified with a higher-level overview or example before diving into math-heavy formulations.

Significance: The concept of pluralistic dominance fills a critical gap in imitation learning by addressing the preservation of diverse high-quality behaviors rather than average performance. Potential for significant impact in applications requiring robustness or multimodal imitation, such as robotics, autonomous driving, and human-agent interaction.
Generalization beyond hand-crafted reward features is mentioned as future work but limits current real-world applicability. Experimental validation is compelling but relatively limited in scale; larger-scale benchmarks would strengthen the claim of practical impact.

Originality: Introduction of pluralistic stochastic dominance as a criterion in imitation learning is novel and conceptually rich. The proposed method differentiates itself clearly from previous approaches like GAIL, MaxEnt IRL, and BROIL in both objective and outcomes.
Some connections to mixture model-based methods (e.g., InfoGAIL) could be more deeply explored, especially given their relevance to handling multimodal demonstrations.

---

> ### Author Rebuttal · Authors · 2025-07-30
>
> Thank you for your detailed review and insightful questions.
>
>
> Question #1–Cost representation learning: Learned cost features could immediately be fine-tuned (e.g., adversarially) since our objective is differentiable with respect to cost feature model parameters. Using stochastic dominance as the basis for representation learning more broadly is of great interest, but beyond the scope of the current work.
>
> Question #2–Computational hardness: We avoid the computational challenges of the original optimization (5) using the relaxation in (6).
>
> Question #3–β choice: Larger β parameters encourage the algorithm to try to outperform the demonstrations more. Appropriate parameter choice depends on the scale of the feature values, the range of possible feature values in the environment, and the quality of the demonstrations. For example, if β is small and demonstrations are very suboptimal, the training loss can be driven to zero without “pushing” the learned policy towards the Pareto frontier. From this perspective, β should be adapted so that each cost feature contributes a balanced, non-zero amount to the overall training loss.
>
> Question #4. The optimal transport solution of PSD generally renders latent variables unnecessary. We view this as a significant advantage of PSD, since modeling this latent structure is often challenging. Under PSD, demonstrations could be split into separate batches to decrease the optimal transport complexity if latent task variables could be reliably inferred.
>
> Question #5. Like reward-based (stochastic) policy optimization, our approach suffers from the high variances of each learning step, but the optimization generally trends towards improvement over time, and the model with the best training performance during training is generally reliable.
>
> We plan to incorporate this discussion into our revision and would appreciate any additional questions or suggestions.

---

> > ### Comment · Reviewer_EQ92 · 2025-08-06
> >
> > Thank you for addressing my questions. I appreciate the clarifications provided.

---

### Official Review · Reviewer_3HFR · 2025-07-02

**Clarity:** 2
**Significance:** 3
**Originality:** 4
**Rating:** 5
**Confidence:** 2

**Summary:**

The authors propose a method for imitation learning based on stochastic dominance over the distribution of possible rewards linearly induced by the demonstrations. To do so, the authors use an optimal transport relaxation. The authors evaluate their approach on a lava world, point bot and reacher.

**Questions:**

Question:
- Several of the inverse RL baselines rely on interaction with the environment. My understanding from the paper is that PSD does not require further data collection. Would allowing interaction with the environment allow for stronger theoretical guarantees?

Score increase:
- More extensive experimental analysis, particularly on a less toy example where multiple reward objectives are desirable.

**Ethical Concerns:**

["NO or VERY MINOR ethics concerns only"]

**Final Justification:**

I stand by my original recommendation of accept. While I do think there is further opportunity to strengthen the paper with additional experiments, I believe the paper is theoretically grounded and offers an interesting new approach to the field. This is consistent with the reviews from the other reviewers.

**Limitations:**

Yes.

**Paper Formatting Concerns:**

No concerns.

**Quality:**

3

**Strengths And Weaknesses:**

Strengths
- The proposed approach relies on a unique approach and outperforming the demonstrators across a wide range of reward functions is a powerful result with solid theoretical grounding.

Weaknesses
- One concern I had was the gap between the proposed motivation of the paper and the experiments. Generally, if we are assuming a mixture of behavior in our demonstrations, then why not simply match the distribution of demonstrations (e.g., with the very popular diffusion policy approach (Chi et al., 2023))? While there are practical cases where we might be interested in multiple reward functions (the authors mention risk-sensitivities), the experiments do not examine these settings. This is of concern, given that the reward function is assumed to be linear with the features. As such, it’s unclear to me whether the paper is actually more effective than the baselines at the types of problems it's attempting to address, rather than just toy constructions with randomly sampled rewards.
- It’s fairly clear but worth mentioning that the proposed approach introduces a more complex optimization problem than in typical imitation learning. Consequently, it’s unlikely to have immediate practical implications.

---

> ### Author Rebuttal · Authors · 2025-07-30
>
> Thank you for your positive review and insightful points/questions.
>
> Weakness #1–Matching behavior mixtures and diffusion policy: A major trade-off between diffusion-based approaches and PSD is that by requiring cost features, PSD provides performance guarantees over the corresponding space of cost functions. The denoising objectives of diffusion models aim to match demonstrations, including replicating any consistent suboptimalities of the demonstrator. This prevents stochastic dominance from being achieved.
>
> Note that under the perspective of PSD, the distinct sets of demonstrations in Figure 8a (ignoring/avoiding the obstacle) and different demonstrated modes more generally correspond to different reward functions, rather than being (nearly) “tied” in reward in a much more complex reward function (under the energy-based policy model perspective).
>
> Weakness #2–Complexity/practicality: We believe that the added computational complexity of solving an optimal transport problem to match rollouts with demonstrations is not very significant for existing imitation learning pipelines. However, our requirement of many training demonstrations in identical settings/tasks is a significant practical challenge in domains like autonomous driving in which only a single “real-world” demonstration is typically available.
>
> Question #1–Online/offline learning: To clarify, we employ a candidate set of trajectories in the fully realizable experiments and do not require any additional environment interaction. However, for the policy model optimization, we apply Algorithm 1 with on-policy samples from the imitator and a fixed set of demonstrations in our experiments. Offline/off-policy extensions when further interactions with a simulator/environment are expensive are possible by employing importance weighting in Step 1 of the algorithm. We do not explore this experimentally in this paper though. We expect that our method can be made more data efficient in the interactive imitation learning setting (e.g., with the learner able to query the demonstrator/expert in learner-chosen states/histories) by better focusing on the demonstrator’s trajectory distribution nearer to the Pareto frontier.
>
> We plan to integrate these clarifications into our revision and would appreciate any additional questions or suggestions.

---

> > ### Comment · Reviewer_3HFR · 2025-08-04
> >
> > Thank you for the clarifications. I am happy to continue to reccomend acceptance of the paper.

---

### Decision · Program_Chairs · 2025-09-17

**Decision:**

Accept (spotlight)

**Comment:**

The authors proposed a method for imitation learning based on stochastic dominance over the distribution of possible rewards linearly induced by the demonstrations. To do so, the authors use an optimal transport relaxation. The authors evaluate their approach on a lava world, point bot and reacher.

The reviewers found that the paper has made significant contributions. In particular, they found that the shift from matching moments to dominating distributions is a significant conceptual contribution to the field. The paper is also strongly theoretically grounded. PSD is rigorously defined and connected to prior work in optimal transport and stochastic dominance. Derivation of tractable approximations via convex optimization is well-justified and novel. The experiments are also strong: the proposed approach outperforms the demonstrators across a wide range of reward functions. Finally, the paper is clearly written and well-organized,

The reviewers would encourage the authors to expand the applicability of the proposed framework: currently, the framework is reliant on a set of pre-defined, hand-engineered reward features.